# Identification of a Novel Frameshift Variant of *ARR3* Related to X-Linked Female-Limited Early-Onset High Myopia and Study on the Effect of X Chromosome Inactivation on the Myopia Severity

**DOI:** 10.3390/jcm12030835

**Published:** 2023-01-20

**Authors:** Xuan Xiao, Jingmin Yang, Ying Li, Hongxia Yang, Yijian Zhu, Lianbing Li, Qinlinglan Zhou, Daru Lu, Ting Chen, Yafei Tian

**Affiliations:** 1Department of Eye Center, Renmin Hospital of Wuhan University, Wuhan 430060, China; 2NHC Key Laboratory of Birth Defects and Reproductive Health, Chongqing Population and Family Planning Science and Technology Research Institute, Chongqing 401120, China; 3State Key Laboratory of Genetic Engineering, School of Life Sciences, Fudan University, Shanghai 200438, China; 4Shanghai WeHealth BioMedical Technology Co., Ltd., Shanghai 201210, China

**Keywords:** *ARR3*, early-onset high myopia, X chromosome inactivation, female-limited, X-arrestin, arrestin3, preimplantation and prenatal genetic testing

## Abstract

X-linked myopia 26 (Myopia 26, MIM #301010), which is caused by the variants of *ARR3* (MIM *301770), is characterized by female-limited early-onset high myopia (eo-HM). Clinical characteristics include a tigroid appearance in the fundus and a temporal crescent of the optic nerve head. At present, the limited literature on eo-HM caused by *ARR3* mutations shows that its inheritance mode is complex, which brings certain difficulties to pre-pregnancy genetic counseling, pre-implantation genetic diagnosis, and prenatal diagnosis. Here, we investigated the genetic underpinning of a Chinese family with eo-HM. Whole exome sequencing of the proband revealed a novel frameshift mutation in *ARR3* (NM_004312, exon10, c.666delC, p. Asn222LysfsTer22). Although the mode of inheritance of the eo-HM family fits the X-linked pattern of *ARR3*, the phenotypes of three patients deviate from the typical early-onset high myopia. Through X-chromosome inactivation experiments, the patient’s different phenotypes can be precisely explained. In addition, this study not only enhanced the correlation between *ARR3* and early-onset high myopia but also provided explanations for different phenotypes, which may inspire follow-up studies. Our results enrich the knowledge of the variant spectrum in *ARR3* and provide critical information for preimplantation and prenatal genetic testing, diagnosis, and counseling.

## 1. Introduction

High myopia is a serious condition of myopia, which is defined as a refractive error ≤ −6.00 diopters (D) or axial length (AL) > 26 mm [1,2,3]. It is one of the leading causes of blindness [1,4,5]. The prevalence is 1~6% in Asian populations and is much higher in high school and college students [6,7]. However, the exact mechanism of myopia development remains unclear [3,8,9]. Both environmental and genetic factors can affect the development of high myopia [3]. Early-onset high myopia with minimum influence from the environment, which starts in preschool years, is considered a unique resource for the identification of high myopia genes [8,10,11,12]. 

X-linked myopia 26 caused by the variants of *ARR3* is characterized by female-limited early-onset high myopia. Clinical characteristics include a tigroid appearance in the fundus and a temporal crescent of the optic nerve head [10]. The arrestin protein encoded by *ARR3* is highly expressed in the retina and localized in the cone photoreceptors [13]. It binds to activated phosphorylated G-protein-coupled receptors, leading to the termination of G-protein-coupled receptor signaling, thus playing a key role in retinal signal transduction.

Some genes are expressed differently in male and female tissues, especially in organs related to sex, such as the breast and prostate. More than a third of genes in humans have sex differences in at least one tissue, and these sex differences can be involved in a variety of diseases, such as immune responses and tumors [14]. The expression of the genes mapped on the sex chromosome and all epigenetic interactions can affect disease development [15]. The sex differences in human disease are usually attributed to gender-specific life experiences and sex hormones that influence the function of susceptible genes [16,17,18]. X-chromosome inactivation is an important epigenetic mechanism for balancing gene dosage between XY males and XX females in eutherian mammals [19]. However, the different impact of X-chromosome inactivation as a major cause of sex-determined disease manifestation is often underestimated [16,20]. The effect of random X-chromosome inactivation on X-related diseases can be observed in many sex-linked diseases [21,22,23,24]. Males are usually affected by X-linked pathogenic mutations, while females often have a milder phenotype even if they carry the same mutation, unless they are homozygous for the deleterious allele, or it is lethal for males [16]. 

In this study, we discovered a novel mutation in *ARR3*, providing further proof of the link between *ARR3* and X-linked female-limited eo-HM. Unlike the previously published studies [10,25,26], not all of our patients have early-onset high myopia. There are two patients having milder phenotypes without the characteristic early onset of the disease. Thus, even though they carry the same mutation, the phenotypes of patients are different. Through the X-chromosome inactivation experiment, it can be observed that the different phenotypes of patients are consistent with the proportion of mutant X-chromosome inactivation, which can well explain the different phenotypes of patients with the same mutation. Although the mechanism of *ARR3* mutation and the accompanying X- chromosome inactivation shift is unknown, our research and literature suggest that HM caused by *ARR3* LOF mutation is more common in women. HM might influence female reproductive health, especially retinal thickness in the second and third trimesters of pregnancy [27] and delivery mode selection [28]. The limited literature on X-linked myopia 26 caused by *ARR3* mutations shows that its inheritance mode is complex and involves X- inactivation deviation, which brings certain difficulties to pre-pregnancy genetic counseling, pre-implantation genetic diagnosis, and prenatal diagnosis. This study, therefore, provides data that triggers a re-thinking of female-limited early-onset high myopia. When studying sexually linked diseases, especially X-linked sex-linked diseases, the influence of epigenetic regulation such as X-chromosome inactivation should be considered in addition to the influence of genetic mutations.

## 2. Materials and Methods

### 2.1. Study Participants

All patients in this study were recruited by the Department of Ophthalmology, Renmin Hospital of Wuhan University, Hubei Province, China. After obtaining the written informed consent of all patients or their guardians, we collected the genomic DNA of peripheral venous blood leukocytes and the clinical data of all participants. Inclusion criteria of high myopia: refractive error ≤ −6.00 diopters (D) or axial length (AL) > 26 mm [1]; eo-HM: high myopia before school age or 6~7 years old [12,29].

### 2.2. Whole-Exome Sequencing (WES) 

The process of whole-exome sequencing was conducted by Shanghai We-Health Biomedical Technology Co., Ltd. Genomic DNA was isolated from peripheral blood samples of patients by using a commercial kit (TIANGEN, Beijing, China). The sequencing libraries were prepared and captured by using IDT xGen Exome Research Panel v1.0 (IDT, USA). The quality control standards: average sequencing depth ≥ 100×, 20× coverage ≥ 99%, and 30× coverage ≥ 98.5%. Libraries were sequenced with a high-throughput sequencer (Illumina HiSeq4000, San Diego, CA, USA). Variants detected were initially filtered with multistep bioinformatics analysis. The clean reads were mapped to the human reference genome (hg19/GRCH37) by the BWA software (version number bwa-0.7.17-r1188). Variants with a frequency above 1% in gnomAD (http://gnomad-sg.org/ accessed on 11 December 2022) were discarded. Then, according to the patient’s clinical information or preliminary diagnosis, all possible pathogenic genes for a certain symptom or disease could be searched on OMIM (https://omim.org accessed on 11 December 2022), HPO (https://hpo.jax.org accessed on 11 December 2022), Genereviews (https://www.ncbi.nlm.nih.gov/books/NBK1116/ accessed on 11 December 2022), NCBI-GTR (https://www.ncbi.nlm.nih.gov/gtr/all/?term= accessed on 11 December 2022) and MedGen (https://www.ncbi.nlm.nih.gov/medgen/ accessed on 11 December 2022). Finally, single-nucleotide polymorphism (SNV) and insertion or deletion (InDel) were analyzed by GATK (version 4.2.0.0); copy number variant (CNV) was analyzed by XHMM (version 1.0); and CLAMMS software (version clamms 1.1); and loss of heterozygosity (LOH) was analyzed by H3M2 software (version h3m2-20131219).

### 2.3. Variant Confirmation and Structure Prediction

Based on the patient’s phenotype, genotype, and genetic pattern, we designed primers for the candidate pathogenic variant sites to verify their authenticity. At the same time, we performed Sanger sequencing on the parental candidate sites. The primers for *ARR3*: forward primer: 5′-AGGAAGCAGGGGTTCTAGGT-3′; reverse primer: 5′-CACTCTACCCCCTCAACTGC-3′. The possible impact of the mutation was predicted by using online tools such as Mutation Taster (https://www.mutationtaster.org/ accessed on 11 December 2022) and AutoPVS1 (http://autopvs1.genetics.bgi.com/ accessed on 11 December 2022). Nonsense-mediated mRNA decay (NMD) is the most likely consequence of this mutation. In addition to NMD, truncated protein production is also a possible outcome. The structure predictions of wild-type and mutant proteins were performed using an online protein structure prediction and function annotation software: I-TASSER (https://zhanggroup.org/I-TASSER/ accessed on 11 December 2022). The variants are described according to the Human Genome Variation Society (HGVS) nomenclature system.

### 2.4. X-Chromosome Inactivation Experiment

During embryogenesis, one of the female X chromosomes is transcriptionally inactivated. This equalizes the impact of two X chromosomes in females [30]. Currently, methylation-based assay to quantitatively define XCI status in humans is the most widely used [31]. In our study, this method refers to the study of Allen RC et al. [32]. It utilized methylation-sensitive restriction endonuclease HapII (Thermo Scientific™, ER0511, Waltham, MA, USA), a pair of androgen receptor (AR) gene-specific fluorescent marker primers (forward primer: 5′-FAM-CGTCCAAGACCTACCGAGGA-3′ and reverse primer: 5′-GAACCATCCTCACCCTGCTG-3′), a pair of fluorescent marker primers for amplifying MIC2 SURFACE ANTIGEN (MIC2) (forward primer: 5′-FAM-AGCCGTTCCAGAGAGAAA-3′ and reverse primer: 5′-CCCGCATCCACAGAGTAT-3′) and Taq Plus Master Mix (Vazyme, P211-AA, Nangjing, China). A mass of 200 ng of DNA per sample was digested with 20 U of ThermoFisher HapII enzyme in a 20-μL volume of buffer solution for 16 h at 37 °C. A no-enzyme control digest was also set up for each sample. HapII enzyme was then heat-inactivated by incubation at 65 °C for 20 min. Complete digestion was verified using PCR and agarose gel electrophoresis. The 20-μL PCR reaction system included: 10 μL of 2 × Taq Plus Master Mix, 1 μL of each MIC2 primer, 2 μL of digested DNA template, and 16 μL of H_2_O. PCR cycling conditions were as follows: initial denaturation at 96 °C for 5 min, denaturation at 96 °C for 30 s, annealing at 60 °C for 30 s, extension at 72 °C for 45 s, 35 cycles, and then 72 °C 5 min. Two μL of the amplified product were reserved for 2% agarose gel electrophoresis detection. If there was no 361 bp target band, it was completely digested and could proceed to the next step. Digested and undigested DNA was then amplified in duplicate PCRs with AR and MIC primers using the same PCR reaction system and cycling conditions. The amplified products were confirmed by 2% agarose gel electrophoresis and subjected to capillary electrophoresis on a genetic analyzer (Superyears, Classic116, Nanjing, China), and fluorescence values were obtained. The amplified product was sent to capillary electrophoresis for detection, and the fluorescence value was obtained. The offset ratio could be calculated by the following rules: (a) inactivation ratio of paternal X chromosome = signal intensity of paternal X chromosome after digestion/signal intensity of paternal X chromosome before digestion; (b) maternal X-chromosome inactivation ratio = maternal X-chromosome signal intensity after digestion/maternal X- chromosome signal intensity before digestion. The criteria used in this project are: (1) the inactivation ratio of X chromosomes from paternal or maternal origin < 70% is defined as random inactivation; (2) the inactivation ratio of X chromosomes from paternal or maternal origin > 70% and <90% at the same time are considered to have moderate inactivation deviation; (3) X-chromosome inactivation ratio from paternal or maternal origin > 90% and above is considered to have extreme inactivation deviation. This standard has been used in many related studies [33,34]. 

## 3. Results

### 3.1. Clinical Characterizations

This study included 10 family members, except for one who had passed away; the rest were investigated. Six women had myopia, three of whom had early-onset high myopia. Due to irreconcilable family conflicts, we could only obtain clinical and mutational information from the small group of the proband family (Table 1). The refractive errors of the four people from the family ranged from −1.25 to −9.50 diopter spheres for the left eye and from −0.50 to −7.75 diopter spheres for the right eye. The axial length of the eye globe ranged from 23.91 to 27.55 mm in the left eye and 24.02 to 27.26 mm in the right eye. Ultrasound results showed that the patients’ binocular axis lengthened, which was consistent with the sonographic changes of high myopia. The echo of the posterior wall of both eyes changed, which was considered local tissue edema. Consistent with previous reports, all six affected family members in our study had myopia before school age. After mydriasis, all the patients had typical leopard fundus changes, as demonstrated by color fundus photography (Figure 1), and the axial lengths of some patients were more than 26 mm (Table 1), with no oculopathy or systemic disease. We tried to persuade the relatives of the proband’s mother to participate in this study. However, due to family conflicts, we were unable to obtain blood samples for sequencing and analysis of other family members’ DNA or clinical data. We could only describe the condition of the disease through the patient’s description and infer its inheritance. Interestingly, after we learned the family history of the proband, we found that the proband’s cousin and aunt also have a history of early-onset high myopia, but grandfather and other men do not have early-onset high myopia.

### 3.2. The Mutation in ARR3 Accounts for Early-Onset High Myopia

There are many genes associated with myopia. Therefore, excluding other myopia-related genes is necessary. After whole-exome sequencing of the proband, three related variants associated with myopia were identified. The heterozygous variant (NM_000093, exon5, c.698C>G, p.A233G) in *COL5A1* and heterozygous variant (NM_004727, exon2, c.193C>T, p.R65W) in *SLC24A1* were excluded due to large phenotypic differences and inconsistent inheritance patterns. A heterozygous frameshift deletion mutation (NM_004312, exon10, c.666delC, p. Asn222LysfsTer22) found in *ARR3* caught our attention (Figure 2). This mutation was assumed to disrupt gene function by leading to the complete absence of the gene product by lack of transcription or nonsense-mediated decay of an altered transcript. The online tool MutationTaster gave five possible results: NMD, amino acid sequence changed, frameshift, protein features (might be) affected, and splice site changes. The production of truncated protein is also one of the possible results. The structure predictions of wild-type and mutant proteins were performed using the I-TASSER (Figure 3). Predictions of the three-dimensional structure of the mutant and wild-type ARR3 showed that frameshift mutations can cause changes in the ligand-binding region, allowing cone-blocking proteins to bind to different substances. This mutation is a rare mutation that has not been reported in the literature, and it is not included in the 1000 G (https://www.internationalgenome.org/ accessed on 11 December 2022), ExAC (http://exac.broadinstitute.org accessed on 11 December 2022), gnomAD (http://gnomad-sg.org/ accessed on 11 December 2022), or EVS (https://evs.gs.washington.edu/EVS/ accessed on 11 December 2022). The main clinical manifestations of the disease include severe myopia in early childhood, usually having an onset before 6~7 years of age, and a female-restricted fashion (hemizygous males are not affected). According to the guidelines of the American College of Medical Genetics and Genomics (ACMG), combined with clinical information, this mutation is likely pathologic. The evidence used for rating is PVS1 (the pathogenic mechanism of the *ARR3* gene is loss of function (LOF), and the mutation of current subjects is frameshift deletion mutation, which is predicted to lead to NMD, resulting in gene loss of function) and PM2_Supporting (the variation was rare and not included in the gnomAD database).

From the rating of the mutant, combined with the proband’s clinical symptoms, family history, and other examination results, we finally inferred that *ARR3* mutation (c.666delC, p.Asn222LysfsTer22) might be the cause of the proband’s high myopia. Therefore, we performed a first-generation sequencing verification for this site, and the verification result was consistent with the NGS result. In addition, we also performed a first-generation sequencing analysis of this locus on the proband’s sister and parents (Figure 2). The results showed that the father didn’t have the mutant, while the proband’s sister and mother also had the same heterozygous mutation at this locus, which was a dominant X inheritance. Since all patients in this family are female, this reminded us of a special genetic model—female-restricted X-linkage, which was similar to the research report of Xiao et al. [10].

### 3.3. The Role of X-Chromosome Inactivation in Early-Onset High Myopia

Since the phenotype of the mother and sister is an atypical early-onset high myopia phenotype, it remains unclear whether it can be explained by the female-restricted genetic model. Due to the complexity of the X chromosome, it is possible that this genetic model could also be influenced by epigenetic regulations. For this reason, we conducted X-chromosome inactivation experiments to explore the expression ratio of mutant X chromosomes (Figure 4). Female somatic tissue is a mosaic composed of two types of cells. In some cells, the paternal X chromosome is active, while the maternal X chromosome is active in the remaining cells. Theoretically, the ratio of these two types of cells is 1:1. When this ratio deviates significantly, it is termed X-chromosome inactivation shift. Currently, there is a variety of criteria in the research field for defining X-chromosome inactivation deviation. In our case, the maternal mutant X-chromosome inactivation ratio of *ARR3* was 81.30%, reaching a moderate inactivation shift, which corresponds to the current mild phenotype. The ratio of mutant X-chromosome inactivation in the proband was 24.37%, that is, the proband mainly expressed the mutant chromosome, and it also reached a moderate inactivation shift, thus the corresponding phenotype was also most severe. The sister of the proband has reached high myopia, but not with early onset. The inactivation ratio of her mutant X chromosome is 56.12%, and 43.88% of the mutant X chromosomes are still active, again, matching the phenotype severity. Since this protein is involved in the phosphorylation of key G-protein-coupled receptors and plays an important role in the light signal transduction of the retina, we speculate that insufficient or abnormal expression of this protein will cause abnormal signal transmission, thereby affecting the visual system development.

## 4. Discussion

In recent years, next-generation sequencing technology has been widely used to decipher genetic determinants. To date, at least 17 pathogenic genes that cause non-syndromic high myopia have been discovered [8,35]. These genes include 11 autosomal dominant genes *ZNF6441*, *SCO2*, *SLC39A5*, *CCDC111*, *P4HA2*, *BSG, CPSF1, NDUFAF7*, *TNFRSF21*, *XYLT,* and *DZIP1*; four autosomal recessive genes, *LRPAP1*, *CTSH*, *LEPREL1,* and *LOXL3*; and two X-chromosome genes, *ARR3* and *OPN1LW*. Most of these genes are inherited in an autosomal dominant manner, and most of the mutation types are missense mutations [36]. However, these known genes can only explain the pathogenesis of a small number of cases. The current research confirms the high genetic heterogeneity and complexity of early-onset high myopia. 

To date, there are seven likely pathogenic or pathogenic *ARR3* variants that have been reported or included in the database (Table 2). Xiao et al. reported the *ARR3* gene-related eo-HM and established an X-linked female restriction inheritance model for the first time [10]. In their study, they identified three heterozygous mutations (p.L80P; p.R100X; p.A298D) in *ARR3* which were predicted to be damaging and co-segregated with the disease in three different families. Heterozygous mutations in *ARR3* may cause eo-HM in women, but in men, hemizygous mutations do not cause eo-HM. Noémi Széll reported the first case of a novel mutation (p. Arg72Ter) in *ARR3* causing hereditary eo-HM in a Caucasian family [25]. Combined with previous animal experimental data, they made some inferences through the results of electrophysiology and color vision testing and proposed two hypotheses about the pathogenic mechanism of *ARR3* mutation. However, no further explanations have been provided for the non-early-onset high myopia phenotype and female-restricted pattern. Ralph van Mazijk et al. identified three novel mutations (c.214C>T; c.767+1G>A; c.848delG) in *ARR3* associated with female-restricted eo-HM in three unrelated multigenerational families [26]. In their study, several male relatives had astigmatism and mild myopia, but none showed eo-HM. The pattern of affected females in these families, and possibly mildly affected males, followed a mode of inheritance known as X-linked female limited. From the point of view of the location and type of mutation, the mutation (c.666delC) in our case is somewhat similar to one mutation (c.848delG) described by Ralph van Mazijk. Similar to our patients, the patient’s medical history included female-limited early-onset high myopia. No male relatives were affected by the mutation (c.848delG). Unlike those patients, in this case, our patients show varying degrees of myopia. Different from these studies, the study by Dejian Yuan et al. showed that high myopia caused by *ARR3* may be not X-linked female-limited, and a complicated X-linked inheritance pattern may exist [37]. However, only one of the eight patients with the variant (c.569C>G) was male and the male patient was relatively late-onset. The conclusion that high myopia caused by *ARR3* is not female-limited may need further discussion. In our study, all the patients were female, but not all had early-onset high myopia. Through the X-chromosome inactivation analysis, we explained the different phenotypes of myopia. Thus, this case suggests that, although myopia-26 is an X-linked-dominant female-restricted inheritance model, some women may have a milder phenotype or even no phenotype due to the effect of X-chromosome inactivation.

The arrestin protein encoded by *ARR3* is highly expressed in the retina and localized in the cone photoreceptors [13,38]. This protein binds to activated phosphorylated G-protein-coupled receptors, leading to the termination of G-protein-coupled receptor signaling, thus playing a key role in retinal signal transduction. The novel frameshift mutation of *ARR3* (NM_004312, exon10, c.666delC, p. Asn222LysfsTer22), in our case, has an important impact on protein function. Although NMD is predicted to be the most likely cause of the loss of function of the *ARR3* gene, truncated protein production is also one of the possible outcomes. The predicted 3D structure of *ARR3* demonstrated that the frameshift mutation can cause changes in the ligand-binding regions, so that cone arrestin protein may bind different substances. To summarize, NMD and truncated nonfunctional proteins can cause the mutated *ARR3* gene to lose function.

## 5. Conclusions

In conclusion, our study provides additional evidence to support the conclusion that *ARR3* is one of the causative genes for eo-HM. Further, it provides more evidence for the link between *ARR3* and X-linked female-limited eo-HM. However, due to the lack of complete family genetic data and the relatively small sample size, our research still has some deficiencies. Although it is a small family, there is still sufficient evidence to prove the pathogenicity of the *ARR3* mutation (NM_004312, exon10, c.666delC, p. Asn222LysfsTer22) in our research. It has the significance of scientific research and clinical guidance that cannot be ignored. How the *ARR3* affects eo-HM and its inheritance patterns warrants further investigation. Our results enrich the knowledge of the variant spectrum in *ARR3* and provide critical information for preimplantation and prenatal genetic testing, diagnosis, and counseling.

## Figures and Tables

**Figure 1 jcm-12-00835-f001:**
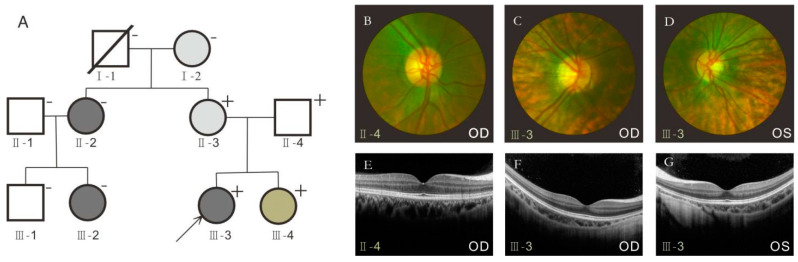
Fundus photographs and optical coherence tomography (OCT) images of control (II-4) and the proband (III-3) in a large family with early-onset high myopia (eo-HM). (**A**) Open squares and solid circles represent normal male and female patients. Compared with the control (**B**), both eyes of the proband showed leopard appearance, peripapillary atrophy, macular thinning, and depigmentation (**C**,**D**). Optical coherence tomography (OCT) image is used as a control (**E**). Foveal flattening and choroidal thinning were shown in both eyes (**F**,**G**). “+” indicates DNA available for WES and direct sequencing; * indicates that the patient mutation was discovered by direct sequencing. “/” represents the dead family member; “↗” indicates the proband; OD, right eye; OS, left eye. Dark gray indicates that the patient has early-onset high myopia; light gray indicates that the patient has moderate or mild myopia, and green indicates that the patient has high myopia but not early-onset myopia.

**Figure 2 jcm-12-00835-f002:**
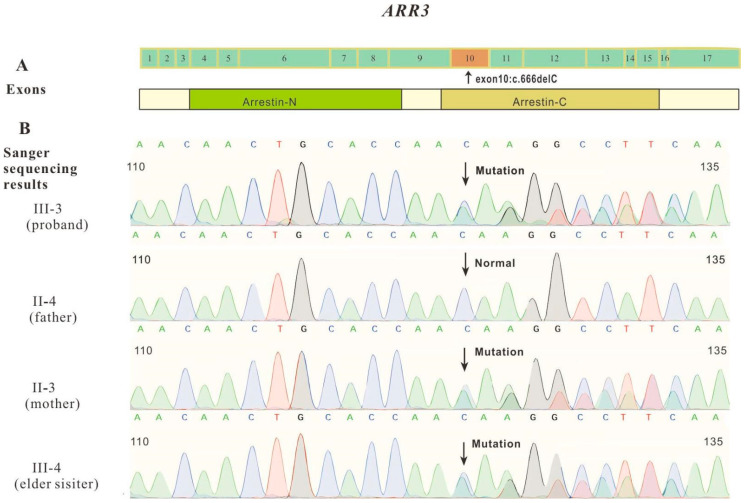
The distribution of mutation (c.666delC) in *ARR3* and Sanger sequencing verification in four family members. The heterozygous mutation in the exon 10 of the *ARR3*, which is located in the arrestin-C domain region. This mutation would cause the protein to change from asparagine to lysine at position 222, and then the frame shifted to produce a stop codon at position 243, which severely affected the protein signal transduction function (**A**). The Sanger sequencing results showed that, except for the father, all three family members had this heterozygous mutation (**B**).

**Figure 3 jcm-12-00835-f003:**
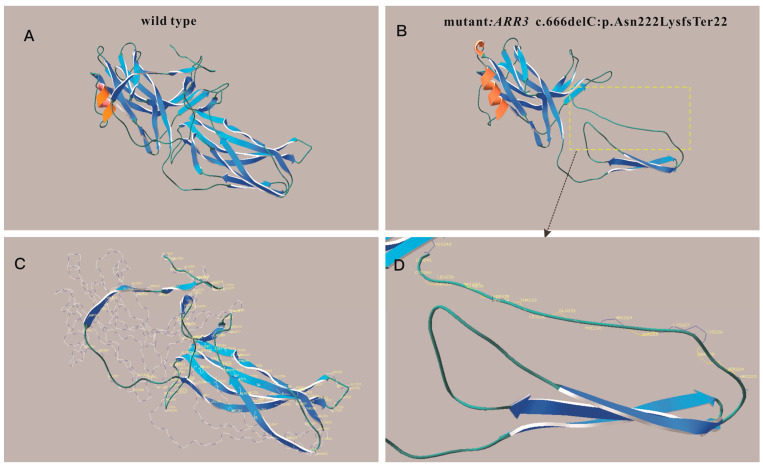
The structure of the proteins predicted by I-TASSER server. The 3D protein model of wild-type (**A**) and mutant protein (**B**). The ribbons (**C**) represent the range of protein deletion caused by the mutation, and the remaining unaffected parts are shown in molecular structure. (**D**) is an enlarged view of (**B**) showing the detailed amino acid changes after position 222.

**Figure 4 jcm-12-00835-f004:**
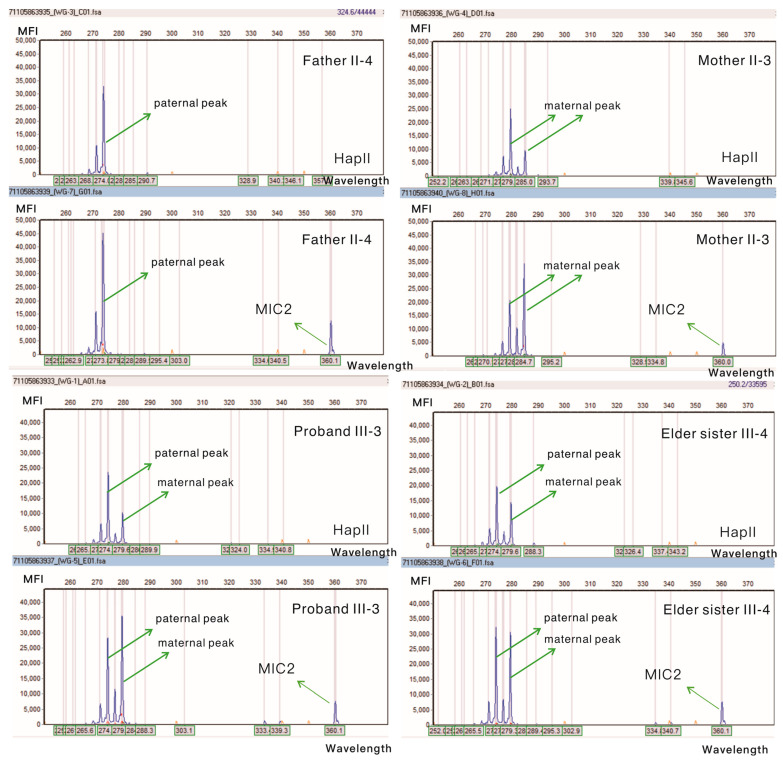
Capillary electrophoresis of products before and after HapII enzyme treatment. HapII: methylation-sensitive restriction endonuclease; MFI: mean fluorescence intensity; by family inference method, 274 is the paternal peak, and 280 and 285 are the maternal peaks. MIC2: control gene: one of the escaped genes from XCI in the females.

**Table 1 jcm-12-00835-t001:** The clinical characteristics in the family with early-onset high myopia.

Subject No.	Age	Gender	Affected Status	Refractive Error (DS)		Axial Length (mm)		Mutations of ARR3
				OD	OS	OD	OS	
II-4	34	Male	U	−0.50	−1.25	24.02	23.91	NO
II-3	34	Female	A	−2.25	−1.75	24.41	24.57	c.666delC: p.Asn222LysfsTer22
III-3	5	Female	A	−7.75	−9.50	25.19	25.32	c.666delC: p.Asn222LysfsTer22
III-4	11	Female	A	−4.50	−5.00	27.26	27.55	c.666delC: p.Asn222LysfsTer22

A, affected; DS, diopter sphere; NO, no mutation; OD, right eye; OS, left eye; U, unaffected.

**Table 2 jcm-12-00835-t002:** Summary of *ARR3* mutations in eo-HM patients.

Ancestry	Gender	cDNA Change	Het/Hemi	Consequence	Protein	Reference
Chinese	Female	c.893C>A	Het	Missense	p.Ala298Asp	[10]
Chinese	Female	c.298C>T	Het	Prem. Stop codon	p.Arg100Ter	
Chinese	Female	c.239T>C	Het	Missense	p.Leu80Pro	
Caucasian	Female	c.214C>T	Het	Prem. Stop codon	p.Arg72Ter	[25,26]
Caucasian	Female	c.767+1G>A	Het	Splicing		[26]
Caucasian	Female	c.848delG	Het	Frameshift	p.Gly283AlafsTer42	
Chinese	Female	c.569C>G	Het	Prem. Stop codon	p.Ser190Ter	[37]
	Male		Hemi			
Chinese	Female	c.666delC	Het	Frameshift	p.Asn222LysfsTer22	This work

Het: heterozygous; Hemi: hemizygote; Prem. Stop codon: premature stop codon.

## Data Availability

The data from this study are not publicly available due to privacy or ethical restrictions but are available on request from the corresponding author.

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
