# Peer review of "Identification of a Novel Frameshift Variant of ARR3 Related to X-Linked Female-Limited Early-Onset High Myopia and Study on the Effect of X Chromosome Inactivation on the Myopia Severity"

_jcm, 2023, doi:10.3390/jcm12030835_

Round 1

Reviewer 1 Report

High myopia is a severe condition of myopia and is one of the leading causes of blindness. Female-limited early-onset high myopia (eo-HM), also called Myopia-26, is a rare monogenic disorder characterized by severe short-sightedness starting in early childhood and progressing to blindness potentially by the middle ages. Genetically, Myopia-26 is thought to be predominantly inherited in a Mendelian manner with one single causative, highly penetrant gene mutation, practically with minimal influence of environment or behavior. This disease paradoxically affects females only, with males being asymptomatic carriers. The ARR3 gene, residing on the X-chromosome and encoding the cone-arrestin, was mutated in all affected patients. Previous studies have revealed several mutated loci of the ARR3 gene in Myopia-26 patients. Clinically, this disease is characterized as early-onset HM before school-age females, a refraction error exceeding or equal to 6.00 dioptres (D) or ocular axial length (AL) >26 mm, a typical tigroid appearance in the fundus, and a temporal crescent of the optic nerve head.

In this manuscript, the authors reported the genetic underpinning of a Chinese family with eo-HM. They revealed a novel frameshift mutation in ARR3 (NM_004312, exon10, c.666delC, p. Asn222LysfsTer22). Their primary findings are that, regardless of the mode of inheritance of the eo-HM family that fits the X-linked pattern of ARR3 mutations, the phenotypes of three patients deviate from the typical eo-HM. These deviations include that not all patients have eo-HM, and two patients have milder phenotypes. They attribute the difference in phenotypes to the proportion of mutant X chromosome inactivation. Accordingly, they concluded that the ARR3 inheritance mode is complex and involves X inactivation deviation. Therefore, the influence of epigenetic regulation should be considered in the study of Myopia-26.

The manuscript's strengths include finding a new ARR3 mutation correlated with Myopia-26-like phenotypes, the variation of the phenotypes, and the association of the phenotypes with X inactivation. However, a weakness of this study is that the family investigated in this report is small. Furthermore, not every patient of this family is included in the clinical and mutational analysis due to “irreconcilable family conflicts.” Therefore, only three female patients and one male control are included for clinical and mutational analysis. The authors should discuss the potential impact of this small sample size on the reliability of the conclusion.

Reviewer 2 Report

The pathologic role of ARR3 mutation in early onset-high myopia (eo-HM) has been described back in 2016 (Xiao et al., Molecular Vision 2016; 22:1257-1266, 2016) where an X-linked, female-limited pattern of inheritance was proposed based on the observation that none of the hemizygous male family members were diagnosed with HM.  Since then a handful of attempts have been made to identify the unique mutations in the ARR3 gene in the females. In the current report the authors have investigated the genetic underpinning of a Chinese family with eo-HM. The whole exome sequencing-based approach identified a novel frameshift mutation (NM_004312, exon10, c.666delC, p. Asn222LysfsTer22) in the proband. The authors concluded that this study provides additional information on ARR3 mutations in relation to eo-HM in understanding the X-linked inheritance.

1.       Overall this study does not appear to be very novel and it does not provide any mechanistic explanations as to how the frameshift mutation in ARR3 is functionally linked to eo-HM.

2.       Why three of the female family members do not show any eo-HM phenotype despite the fact that they all carry the same frameshift mutations in the ARR3 gene.

3.       Citation and description of figure 3 is nowhere found in the manuscript text.

Round 2

Reviewer 2 Report

The responses provided by the authors addressing my comments and concerns are satisfactory. The revised manuscript appears much improved.